# ALDEN: REINFORCEMENT LEARNING FOR ACTIVE NAVIGATION AND EVIDENCE GATHERING IN LONG DOCUMENTS

## ABSTRACT

Vision–language models (VLMs) excel at interpreting text-rich images but struggle with long, visually complex documents that demand analysis and integration of information spread across multiple pages. Existing approaches typically rely on fixed reasoning templates or rigid pipelines, which force VLMs into a passive role and hinder both efficiency and generalization. We present **A**ctive **L**ong-**D**ocum**E**nt **N**avigation (ALDEN), a multi-turn reinforcement learning framework that fine-tunes VLMs as interactive agents capable of actively navigating long, visually rich documents. ALDEN introduces a novel `fetch` action that directly accesses the page by index, complementing the classic `search` action and better exploiting document structure. For dense process supervision and efficient training, we propose a rule-based cross-level reward that provides both turn- and token-level signals. To address the empirically observed training instability caused by numerous visual tokens from long documents, we further propose a visual-semantic anchoring mechanism that applies a dual-path KL-divergence constraint to stabilize visual and textual representations separately during training. Trained on a corpus constructed from three open-source datasets, ALDEN achieves state-of-the-art performance on five long-document benchmarks. Overall, ALDEN marks a step beyond passive document reading toward agents that autonomously navigate and reason across long, visually rich documents, offering a robust path to more accurate and efficient long-document understanding. All code and datasets will be released on to support future research.

## 1 INTRODUCTION

Visually rich documents (VRDs) serve as primary vehicles for storing and communicating structured knowledge in real-world applications. Unlike plain text, these documents combine different modalities, including text, tables, and figures, embedded in human-friendly layouts that encode semantic relationships. Effectively understanding such documents requires not only extracting textual content but also reasoning over their visual and structural organization. This has given rise to the task of visually rich document understanding (VRDU) (Wang et al., 2023; Ding et al., 2022) which aims to develop systems to automatically analyze VRDs and answer user queries, underpining various practical applications (Liang et al., 2024; Rombach & Fettke, 2024)

Despite recent progress of vision-language models (VLMs) on single-page or short documents (Xie et al., 2024; Lv et al., 2023; Hu et al., 2024), real-world long documents spanning dozens or even hundreds of pages remain highly challenging. Feeding entire documents into a model's context is computationally expensive and introduces substantial noise, making it difficult for VLMs to focus on relevant pages (Cho et al., 2024). A more scalable alternative is to have the VLM reason only over semantically relevant pages retrieved by a multimodal retriever (Faysse et al., 2025), following the retrieval-augmented generation (RAG) paradigm (Cho et al., 2024; Chen et al., 2025a). Recent work has extended this idea by building prompting-based pipeline in which VLMs passively perform predefined subtasks such as query reformulation or retrieval analysis within fixed workflows (Han et al., 2025; Wang et al., 2025b). While effective, these systems rely on static reasoning patterns and rigid workflows, limiting their ability to generalize or adapt strategies to diverse user queries. This motivates shifting the research focus to the **Agentic VRDU** (A-VRDU) task, which requires

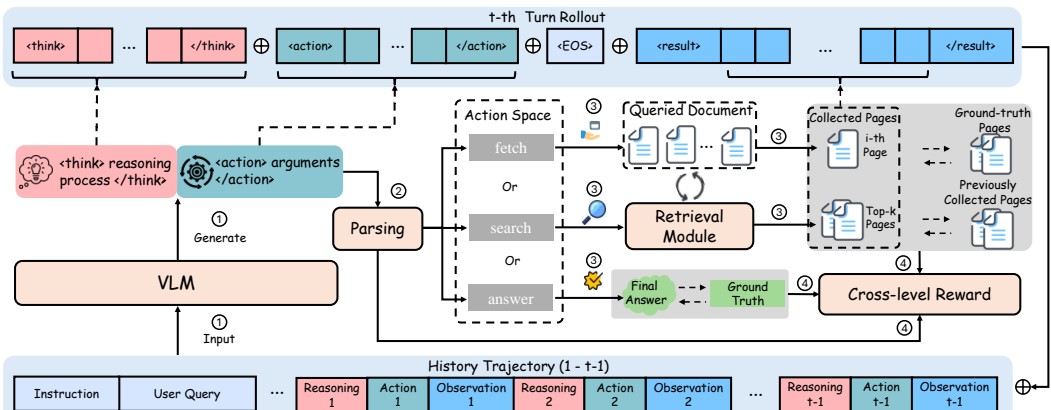

Figure 1: Overview of the rollout process. At each turn: (1) the VLM generates a response conditioned on the dialogue history; (2) the response is parsed into an action (`search`, `fetch`, or `answer`); (3) the action is executed, where `search` or `fetch` collect document pages and `answer` terminates the process; and (4) the cross-level reward function assigns rewards based on execution outcomes and parsing results.

the model to act as an agent that can actively navigate and reason over long documents to deliver accurate and adaptive question answering beyond fixed RAG pipelines.

Recent studies (Chen et al., 2025b; Jin et al., 2025; Song et al., 2025) show that modeling search as an action and optimizing the workflow with outcome-based RL yields more generalizable agents that can actively gather information from external databases, offering a promising direction for the open problem of A-VRDU. However, extending this framework to fine-tune VLMs for A-VRDU poses unique challenges. User queries often reference specific documents, page numbers, or require reasoning across consecutive pages, where generic semantic retrieval is inefficient. Moreover, document-level information gathering typically demands multi-turn interaction with retrieval models, where sparse and delayed outcome-based rewards fail to reinforce helpful intermediate steps or discourage redundant actions. A further challenge arises from the high-dimensional visual inputs. We empirically observe that fully masking the visual tokens when computing the policy gradient, as in existing approaches, leads to unstable training dynamics and can even cause collapse.

These limitations motivate our framework, **A**ctive **L**ong-**D**ocum**E**nt **N**avigation (ALDEN), a multi-turn RL framework that trains VLMs as interactive agents for navigation in long, visually-rich documents. The overall reasoning-action rollout of ALDEN is illustrated in Fig. 1. ALDEN expands the action space by introducing the `fetch` action, which enables direct page-index access to complement search-based retrieval and efficiently handle diverse queries. We incorporate a *cross-level reward function* as opposed to the sparse outcome-based reward typically used, which integrates rule-based turn-level supervision with a token-level repetition penalty to provide fine-grained process supervision, encouraging informative evidence collection while discouraging repeated query formulations. Finally, ALDEN incorporates a *visual semantic anchoring* mechanism, which constrains the hidden states of generated and visual tokens separately during training to preserve the grounding of visual-token representations and improve overall training robustness.

We build a training corpus from DUDE (Van Landeghem et al., 2023), MPDocVQA (Tito et al., 2023b), and SlideVQA (Tanaka et al., 2023b) to train an A-VRDU agent with ALDEN and evaluate it on five benchmarks. Experimental results show that ALDEN achieves state-of-the-art performance over strong baselines and demonstrates the effectiveness of its key components. Overall, the A-VRDU task establishes a new paradigm for processing practical, lengthy VRDs, shifting from passive document reading to autonomous navigation and reasoning. ALDEN's strong results validate this paradigm and provide guidance for building efficient, robust A-VRDU agents from VLMs.

Overall, our main contribution can be summarized as follows:

- We propose the agentic visually-rich document understanding (A-VRDU) task that aims to develop agents that can actively navigate and reason over long visually-rich documents.
- To perform the A-VRDU task, we introduce **ALDEN**, a multi-turn RL framework with three key components: an expanded action space featuring a novel `fetch` action, a cross-level reward

function, and a visual semantic anchoring mechanism, which together enable efficient and robust training.

- We construct a training corpus for training the A-VRDU agent and conduct extensive experiments on five commonly used VRDU benchmarks, showing that ALDEN significantly outperforms the strongest baseline, improving the answer accuracy by 9.14% on average.

## 2 RELATED WORK

### 2.1 VISUALLY-RICH DOCUMENTS UNDERSTANDING

Recent VLMs that process document images directly without OCR (Hu et al., 2024; Xie et al., 2024; Feng et al., 2024; Liu et al., 2024b) have shown strong performance on single-page or short-document benchmarks (Mathew et al., 2021; Masry et al., 2022; Mathew et al., 2022). In contrast, real-world documents often span dozens or hundreds of pages, requiring reasoning across dispersed text, tables, and figures (Deng et al., 2024; Ma et al., 2024b). Extending context length to encode entire documents (Tito et al., 2023b; Blau et al., 2024) is computationally prohibitive and introduces noise, while semantic retrieval provides a more scalable way to focus on relevant pages (Chen et al., 2025b; Jin et al., 2025; Song et al., 2025). However, existing retrieval-based methods largely rely on prompting-based workflows (Han et al., 2025; Wang et al., 2025b), which are static and brittle. In contrast, we study A-VRDU task, and propose to fine-tune VLMs with RL, enabling them to serve as VRDU agents capable of active, multi-step retrieval and reasoning.

### 2.2 RL TRAINING FOR LLMS/VLMS

RL was introduced to LLM fine-tuning by Ouyang et al. (2022); Ziegler et al. (2019) through reinforcement learning from human feedback (RLHF), where a learned reward model guides the RL-based tuning of the policy LLM typically via the Proximal Policy Optimization (PPO) algorithm (Schulman et al., 2017). Recently, RL with verifiable outcome-based rewards (RLVR) (Shao et al., 2024) further demonstrates impressive effect in inducing sophisticated reasoning ability in LLMs. Building on this progress, several recent studies integrate RL with retrieval-augmented generation (RAG), fine-tuning LLMs as agents that actively gather evidence through retrieval and reason over it (Jin et al., 2025; Song et al., 2025). However, extending these methods to the A-VRDU task remains largely unexplored. Unlike open-domain retrieval, VRDU requires exploiting explicit document structure (e.g., page indices), denser supervision to guide multi-turn navigation, and stability against the large number of unconstrained visual tokens introduced by high-resolution document pages, motivating new RL frameworks tailored for this task.

## 3 PRELIMINARIES

### 3.1 PROBLEM FORMULATION

In the A-VRDU task, a user query $q_u$ is paired with a document $\mathcal{D} = (p_1, p_2, \cdots, p_{|\mathcal{D}|})$ that can only be accessed through specific ways, where $p_i$ denotes the $i$-th page and $|\mathcal{D}|$ the total number of pages. The goal is to build an agent that can actively analyzes available information, decides whether and how to collect additional pages from the document, and ultimately generates a final answer $y'$ based on the collected evidence. This sequential decision-making process can be naturally formulated as a Markov Decision Process (MDP) (Bellman, 1957). Formally, at each turn $t$, the agent generates an action $a_t$ from the action space $\mathcal{A}$. Upon executing the action, the document returns a visual observation $o_t \in \mathcal{O}$ (i.e., a page image) and a scalar reward $r_t \in \mathbb{R}$, which reflects the action's utility in acquiring useful evidence or answering the query. The state $s_t$ is defined as the interaction history up to turn $t$, given by $s_t = [x, a_1, o_1, \cdots, a_{t-1}, o_{t-1}]$, where $x$ denotes the initial prompt constructed from the query and task instructions. The agent's objective is to maximize the expected cumulative reward $\sum_{t=1}^{T} \gamma^t r(s_t, a_t)$, where $T$ is the maximum number of interaction turns per episode, $\gamma$ denotes the discount factor.

### 3.2 PROXIMAL POLICY OPTIMIZATION FOR FINE-TUNING LLMS

PPO algorithm is an actor-critic RL algorithm that has been widely used in RLHF to fine-tune language models toward task-specific preferences. In the classical RLHF setup, the problem is typically modeled as a contextual bandit, where each episode involves a single interaction step. Formally, given an input prompt $x$, the LLM auto-regressively generates a variable-length token sequence $(a_1^1, \cdots, a_1^L) \in \mathcal{V}^L$ as a single action $a_1$ where $\mathcal{V}$ denotes the vocabulary and $L$ is the sequence length. A scalar reward $r_1$ is assigned to the action by a learned reward model. Since LLMs operate token-by-token, PPO is actually applied at the token level by treating each token $a_1^i \in \mathcal{V}$ as an action, with state $s_1^i = (x, a_1^1, \ldots, a_1^{i-1})$ defined as the prompt concatenated with the partial response. To propagate the turn-level reward $r_1$ to individual tokens, a token-level reward signal is assigned as

$$r_1^i = \begin{cases} r_1 - \beta \cdot \mathrm{KL}[\pi_\theta(a_1^i|s_1^i)||\pi_{\mathrm{ref}}(a_1^i|s_1^i)], & \text{if } i = L \\ - \beta \cdot \mathrm{KL}[\pi_\theta(a_1^i|s_1^i)||\pi_{\mathrm{ref}}(a_1^i|s_1^i)], & \text{otherwise} \end{cases} \tag{1}$$

where $\pi_{\mathrm{ref}}$ is the reference model (e.g., a frozen copy of the pre-trained LLM), the $\mathrm{KL}(\cdot)$ term acts as a penalty to prevent the policy from drifting too far from the reference model, $\beta$ is the hyperparameter to control the weight of the KL divergence penalty. In addition to the policy $\pi_\theta$, a value function $V_\phi(s_1^i)$ is trained to predict the expected return at each token position. Generalized Advantage Estimation (GAE) (Schulman et al., 2015) is generally used to calculate the advantage of each token-level action:

$$A_1^i = \sum_{k=i}^{L} (\gamma_{\mathrm{token}}\lambda_{\mathrm{token}})^{k-i}\delta_k, \quad \delta_k = r_1^k + \gamma_{\mathrm{token}}V_\phi(s_1^{k+1}) - V_\phi(s_1^k) \tag{2}$$

where $\lambda \in [0, 1]$ is a hyperparameter to balance the estimation bias and variance. The value function is then optimized by minimizing the mean squared error between predicted values and GAE-estimated target values $\hat{V}_1^i = A_1^i + V_\phi(s_1^i)$. The LLM is finally optimized by maximizing the following surrogate objective:

$$\mathcal{L}_{\mathrm{policy}} = \mathbb{E}_x[\sum_{i=1}^{L}[\min\left[\frac{\pi_\theta(a_1^i|s_1^i)}{\pi_{\mathrm{old}}(a_1^i|s_1^i)}A_1^i, \mathrm{clip}\left(\frac{\pi_\theta(a_1^i|s_1^i)}{\pi_{\mathrm{old}}(a_1^i|s_1^i)}, 1-\epsilon, 1+\epsilon\right)A_1^i\right]]] \tag{3}$$

where $\pi_\theta$ and $\pi_{\mathrm{old}}$ are the current and old policy models, $\epsilon$ is a clipping-related hyper-parameter introduced in PPO for stabilizing training. The single-turn PPO framework propagates only immediate rewards to tokens, neglecting each action's contribution to final task completion and fine-grained token supervision. We next describe how we adapt it for long-horizon, multi-turn interaction in the A-VRDU task.

## 4 METHODOLOGY

We propose **A**ctive **L**ong-**D**ocum**E**nt **N**avigation (ALDEN), a reinforcement learning framework for training VLMs as interactive agents that can actively navigate and reason over long, visually rich documents by operating in a multi-turn reasoning–action loop, incrementally collecting evidence pages until a question can be confidently answered. To this end, ALDEN introduces three key components. **(i) Expanded action space:** the agent is equipped with both a semantic `search` action for retrieving relevant pages and a novel `fetch` action for direct page access, enabling flexible exploitation of document structure (§4.1). **(ii) Cross-level reward function:** supervision is provided jointly at the turn level and the token level, guiding the agent toward effective evidence collection and accurate answer generation (§4.2). **(iii) Visual semantic anchoring:** to stabilize RL training, ALDEN constrains the hidden-state evolution of generated and visual tokens respectively, mitigating drift and preserving semantic grounding during optimization (§4.3). The overall RL training pipeline of ALDEN is illustrated in Fig. 2 and Alg. 1.

### 4.1 EXPANDED ACTION SPACE

In Agentic VRDU, agents must flexibly access information that may be referenced either semantically or structurally. Relying solely on semantic retrieval is often insufficient: while it works for

Figure 2: Overview of RL training in ALDEN. The policy model generates multi-turn trajectories, which are scored by a **cross-level reward function** and a **value model**. **Turn-level GAE** integrates future rewards to update the cross-level reward, and **token-level GAE** produces advantages for policy updates. A **reference model** supplies logits for both generated and visual tokens, which the **visual semantic anchoring** mechanism uses to constrain hidden-state evolution during optimization.

open-ended queries, it cannot efficiently resolve explicit page references (e.g.,"see page 12") or reasoning steps that span consecutive pages. To address this, ALDEN augments the standard `search` operation with a complementary `fetch` action, which enables direct page-index access and better exploits the inherent structure of documents. The action space thus consists of three options, each expressed in a structured format that combines free-form reasoning with explicit actions:

- **Search** — `<think>...</think><search>...</search>`
  Generates a reasoning trace within the `<think>` tags followed by a semantic query enclosed within the `<search>` tags. An external retrieval module returns a ranked list of pages relevant to the current query using semantic similarity. This action is effective for open-ended queries where relevant content is not explicitly referenced.
- **Fetch** — `<think>...</think><fetch>...</fetch>`
  Similar to search, but the agent specifies a page number within the `<fetch>` tag, enabling direct access to that page without semantic matching. This action is crucial for handling explicit references to page numbers or structured navigation across consecutive pages.
- **Answer** — `<think>...</think><answer>...</answer>`
  Outputs the reasoning trace followed by the final answer. This action terminates the rollout.

Once the action is parsed, the document returns the corresponding page images enclosed within the `<result>` tag. For the `search` action, the associated page numbers are also returned to provide cues of document structure.

## 4.2 CROSS-LEVEL REWARD MODELING

Training agentic VRDU systems requires reward signals that are both structured enough to enforce valid behaviors and fine-grained enough to guide efficient exploration. To this end, ALDEN employs a cross-level reward function that integrates supervision at two complementary levels: turn-level rewards for overall action quality and token-level rewards for local shaping.

**Turn-level Reward.** The immediate turn-level reward $r_t$ is defined as $r_t = f_t + u_t$, where the format reward $f_t$ enforces the response format and the result reward $u_t$ evaluates the quality of the action outcome. The format reward $f_t$ is given by:

$$f_t = \begin{cases} 0, & \text{if the format is correct} \\ -1, & \text{otherwise} \end{cases} \tag{4}$$

Thus, only well-formed responses avoid penalty, ensuring consistent structured outputs across turns. The result reward is defined based on the action type $a_t \in \{\texttt{search}, \texttt{fetch}, \texttt{answer}\}$, the set of page indices collected in the current turn $\mathcal{C}_t = \{c_1, \ldots, c_{|\mathcal{C}_t|}\} \subseteq \{1, \ldots, |\mathcal{D}|\}$, the set of ground-truth page indices $\mathcal{G} = \{g_1, \ldots, g_{|\mathcal{G}|}\} \subseteq \{1, \ldots, |\mathcal{D}|\}$, and the set of previously accessed pages $\mathcal{R} = \bigcup_{k=1}^{t-1} \mathcal{C}_k$.

$$u_t = \mathbb{1}_{a_t=\texttt{answer}} \cdot \text{F1}(y, y') \cdot \alpha + \mathbb{1}_{a_t=\texttt{fetch}} \cdot (f_{idx}(\{c_1\}, \mathcal{G}) - f_{rep}(\mathcal{C}_t, \mathcal{R}) \cdot \eta)$$
$$+ \mathbb{1}_{a_t=\texttt{search}} \cdot (NDCG@m - f_{rep}(\mathcal{C}_t, \mathcal{R}) \cdot \eta) \tag{5}$$

where $\mathbb{1}(\cdot)$ denotes the indicator function, $\alpha > 1$ scales the reward of `answer` as the outcome reward, and $\eta$ controls the weight of the repetition penalty. The term $F1(y, y')$ is the character-level F1 score between the generated answer $y'$ and the ground-truth answer $y$. For `fetch`, $f_{idx}(\{c_1\}, \mathcal{G}) = e^{-\bar{d}(\{c_1\}, \mathcal{G})}$ smoothly rewards fetching pages near the ground-truth pages, where $\bar{d}(i, \mathcal{G}) = \frac{1}{|\mathcal{G}|}\sum_{i=1}^{|\mathcal{G}|}|c_1 - g_i|$. $NDCG@m$ evaluates the ranked list of retrieved pages, providing a fine-grained reward for `search`. For both `fetch` and `search`, $f_{rep}(\mathcal{C}_t, \mathcal{R}) = \frac{|\mathcal{C}_t \cap \mathcal{R}|}{|\mathcal{C}_t|}$ penalizes repeated page collection. To account for long-horizon credit assignment, following Zhou et al. (2024); Wang et al. (2025a), we extend immediate rewards with turn-level GAE,

$$\hat{V}_t = \sum_{k=t}^{T}(\gamma_{\text{turn}}\lambda_{\text{turn}})^{k-t}\delta_k + V_\phi(s_t^L), \quad \delta_k = r_k + \gamma_{\text{turn}}V_\phi(s_{k+1}^L) - V_\phi(s_k^L) \tag{6}$$

where $V_\phi(s_t^L)$ denotes the value predicted at the last token of the $t$-th response, serving as the turn-level value estimate. The resulting $\hat{V}_t$ replaces the raw $r_t$ as the per-turn reward signal to provide a richer learning signal that aligns token-level updates with long-horizon objectives.

**Token-level Reward.** Unlike the `fetch` action, whose argument is a single page number, the `search` action takes a search query composed of multiple tokens. A turn-level repetition penalty cannot identify which tokens are repeated, and thus fails to effectively curb redundant search actions. To address this limitation, we further introduce a token-level penalty applied specifically to the query span of search actions. Starting from the second invocation of search within an episode, we compute the maximum Jaccard similarity between the current query's n-grams and those of all past queries:

$$\text{overlap}_t = \max_{j<t}\frac{|Q_n(q_t) \cap Q_n(q_j)|}{|Q_n(q_t) \cup Q_n(q_j)|} \tag{7}$$

where $Q_n(q)$ denotes the set of n-grams of the query. To distribute this penalty at the token level, we assign per-token weights so that tokens inside repeated n-grams receive proportionally higher penalties. For each token $u$ in the query span $a_t^{query}$, the weight is defined as $w_u = \frac{c_u}{\sum_{v \in a_t^{query}} c_v}$, where $c_u \in \{0, 1, 2, \cdots\}$ counts how many repeated n-grams include token $u$.

Finally, the reward assigned to each generated token $a_t^i$ within turn $t$ is defined by combining turn-level and token-level signals:

$$r_t^i = \begin{cases} \hat{V}_t, & \text{if } i = L \\ -w_i \cdot \text{overlap}_t, & \text{if } t > 1 \text{ and } a_t = \texttt{search} \text{ and } a_t^i \in a_t^{query} \\ 0, & \text{otherwise} \end{cases} \tag{8}$$

This formulation anchors the turn-level objective to the response boundary, while applying localized penalties to redundant query tokens, yielding a unified cross-level reward signal for token-level PPO training. Token-level GAE is then applied to compute advantages for policy updates as in Eq. (2).

## 4.3 VISUAL SEMANTIC ANCHORING

A unique challenge in RL training for A-VRDU stems from the large number of visual tokens in the trajectory introduced by high-resolution document pages. Without explicit constraints on these tokens, we empirically observe pronounced training fluctuations and rapid entropy collapse (Fig. 3). To address this issue, we propose a Visual Semantic Anchoring mechanism that constrains hidden states during policy optimization through dual-path KL regularization. The KL term for textual tokens regularizes the policy distribution against a frozen reference model, stabilizing language generation, while the KL term for visual tokens anchors their hidden states to the reference model, preserving semantic grounding and preventing drift. Formally, we define

$$\mathcal{L}_{\text{policy}} = \mathbb{E}_x\Big[\frac{1}{T}\sum_{t=1}^{T}\Big[\frac{1}{L}\sum_{i=1}^{L}[\min\Big[\frac{\pi_\theta(a_t^i|s_t^i)}{\pi_{\text{old}}(a_t^i|s_t^i)}A_t^i, \text{clip}\Big(\frac{\pi_\theta(a_t^i|s_t^i)}{\pi_{\text{old}}(a_t^i|s_t^i)}, 1-\epsilon, 1+\epsilon\Big)A_t^i\Big]$$

$$+ \beta_{\text{gen}}KL[\pi_\theta(a_t^i|s_t^i)||\pi_{\text{ref}}(a_t^i|s_t^i)]] + \frac{1}{H}\sum_{j=1}^{H}\beta_{\text{obs}}KL[\pi_\theta(o_t^j|o_t^{<j}, a_t, s_t)||\pi_{\text{ref}}(o_t^j|o_t^{<j}, a_t, s_t)]]] \tag{9}$$

where $H$ denotes the number of visual tokens. $\beta_{\text{gen}}$ and $\beta_{\text{obs}}$ are independent coefficients. In practice, we set $\beta_{\text{obs}} > \beta_{\text{gen}}$ to tightly regularize the much larger observation-token set while allowing more flexibility for generated tokens to adapt to the task.

## 5 EXPERIMENTS

We conduct experiments on long VRDU benchmarks to (i) compare ALDEN with strong baselines and (ii) assess the contribution of its key components, including expanded action space, cross-level reward, and visual semantic anchoring, to navigation accuracy, answer quality, and training stability. We first outline datasets, baselines, implementation details, and evaluation metrics (§5.1), then present main results (§5.2), followed by ablations (§5.3) and detailed component analyses (§5.4).

### 5.1 EXPERIMENTAL SETUP

**Datasets.** We build the training set by merging and processing three multi-page VRDU datasets: DUDE (Van Landeghem et al., 2023), MPDocVQA (Tito et al., 2023a), and SlideVQA (Tanaka et al., 2023a). We filter out documents with fewer than 10 pages. To enrich query diversity, we use GPT-4o (Hurst et al., 2024) to rewrite part of MPDocVQA, increasing the proportion of page-index–referenced queries in the final training corpus. Detailed statistics of the resulting training set are provided in Tab. 1. The evaluation is conducted mainly on the following VRDU benchmarks: **MM-LongBench** (Ma et al., 2024b), **Long-DocURL** (Deng et al., 2024), **PaperTab** (Hui et al., 2024), **PaperText** (Hui et al., 2024), and **FetaTab** (Hui et al., 2024). To evaluate the `fetch` action, we create DUDE-sub, a DUDE validation

Table 1: Statistics of the training dataset. #GQ and #PQ denote the numbers of general user queries and page-index–referenced queries, respectively.

| Sub-dataset | DUDE | SlideVQA | MPDocVQA |
|---|---|---|---|
| #GQ | 6,943 | 10,615 | 7,992 |
| #PQ | 1,011 | 2 | 4,165 |
| Sum | 7,954 | 10,617 | 12,157 |

subset with 480 general queries and 480 queries containing explicit page references or implicit sequential navigation cues. More details about the dataset can be seen in Appx. A.

**Baselines.** To validate the effectiveness of ALDEN, we compare it with three categories of baselines. (1) **Full-Document Input**: mainstream state-of-the-art VLMs are prompted with the entire document as context to answer user queries. (2) **Visual RAG**: methods that retrieve the most relevant document pages using the user query, including M3DocRAG (Cho et al., 2024), and ReSearch-VL, a Search-only ALDEN variant trained with GRPO using outcome-based rewards adapted from a fully textual method ReSearch (Chen et al., 2025b). (3) **Hybrid RAG**: approaches that augment page images with OCR-extracted text for retrieval and reasoning, including MDocAgent (Han et al., 2025), VidoRAG (Wang et al., 2025b). Detailed baseline configurations can be seen in Appx. B

**Implementation Details.** Both the policy and value models are initialized from Qwen2.5-VL-7B-Instruct (Bai et al., 2025), and all Visual RAG and Hybrid RAG baselines use the same backbone for fairness. During training, we adopt the single-vector retriever vdr-2b-v1 (Ma et al., 2024a) for images and e5-large-v2 (Wang et al., 2022) for text. For evaluation, we also report results with the multi-vector retrievers ColQwen2-v1.0 (ColQwen) (Faysse et al., 2025) for images and ColBERT-v2.0 (ColBERT) (Santhanam et al., 2021) for text. Unless otherwise noted, each `search` action retrieves the top-1 candidate page, with a maximum of $T = 6$ reasoning–action turns. On average, ALDEN collects 1.87 unique pages per query; hence, single-turn RAG baselines are set to retrieve the top-2 pages for a fair comparison. Further implementation details are provided in Appx. C.

**Evaluation Metrics.** The primary evaluation metric is GPT-4o–judged answer accuracy (**Acc**) on each benchmark. For finer-grained analysis of ALDEN's components, we further assess navigation quality using trajectory-level retrieval recall (**Rec**), precision (**Pre**), F1-score (**F1**), and the number of unique collected pages (**#UP**). Detailed definitions of these metrics are provided in Appx. D.

### 5.2 MAIN RESULTS

Table 5.2 reports answer accuracy across all baselines. Directly prompting large VLMs with the entire document performs poorly (Acc $< 0.30$), confirming the difficulty of long-document reasoning where irrelevant content overwhelms true evidence. Retrieval-based methods achieve substantially better results. Among Visual RAG approaches, ALDEN with ColQwen attains the highest average accuracy (0.410), surpassing M3DocRAG by 3.2 points. In Hybrid RAG, baselines such as Vi-DoRAG and MDocAgent benefit from textual signals but are limited by fixed reasoning pipelines. ALDEN with hybrid retrievers achieves the best overall performance, exceeding the strongest hy-

Table 2: Answer accuracy comparison on five VRDU benchmarks. † indicates the strongest non-ALDEN baseline used to compute the relative improvement (%). **Bold** indicates the best result per dataset.

| Method | MMLongBench | LongDocUrl | PaperTab | PaperText | FetaTab | Avg |
|---|---|---|---|---|---|---|
| *Full Document Input* | | | | | | |
| SmolVLM-Instruct (Marafioti et al.) | 0.072 | 0.165 | 0.065 | 0.142 | 0.148 | 0.118 |
| Phi-3.5-Vision-Instruct (Abdin et al.) | 0.141 | 0.285 | 0.068 | 0.174 | 0.232 | 0.180 |
| mPLUG-DocOwl2 (Hu et al.) | 0.159 | 0.273 | 0.072 | 0.162 | 0.288 | 0.191 |
| Qwen2-VL-7B-Instruct (Wang et al.) | 0.177 | 0.280 | 0.077 | 0.146 | 0.339 | 0.203 |
| LEOPARD (Jia et al.) | 0.196 | 0.313 | 0.112 | 0.189 | 0.341 | 0.230 |
| Qwen2.5-VL-7B-Instruct (Bai et al.) | 0.221 | 0.375 | 0.131 | 0.265 | 0.336 | 0.265 |
| InternVL3.5-8B-Instruct (Wang et al.) | 0.219 | 0.381 | 0.130 | 0.271 | 0.348 | 0.270 |
| *Visual RAG methods* | | | | | | |
| ReSearch-VL (ColQwen) | 0.274 | 0.384 | 0.150 | 0.295 | 0.406 | 0.302 |
| M3DocRAG (ColQwen)† | 0.330 | 0.464 | 0.201 | 0.350 | 0.547 | 0.378 |
| **ALDEN** (vdr-2b-v1) | 0.335 | 0.513 | 0.201 | 0.342 | 0.542 | 0.386 |
| **ALDEN** (ColQwen) | 0.367 | 0.526 | 0.211 | 0.345 | 0.603 | 0.410 |
| Relative Improvement (%) | 11.21 | 13.36 | 4.98 | -1.43 | 10.23 | 10.81 |
| *Hybrid RAG methods* | | | | | | |
| ViDoRAG (ColQwen + ColBERT) | 0.215 | 0.323 | 0.158 | 0.264 | 0.358 | 0.264 |
| MDocAgent (ColQwen + ColBERT)† | 0.347 | 0.494 | 0.221 | 0.408 | 0.607 | 0.415 |
| **ALDEN** (vdr-2b-v1 + e5-large-v2) | 0.385 | 0.542 | 0.228 | 0.416 | 0.611 | 0.436 |
| **ALDEN** (ColQwen + ColBERT) | **0.392** | **0.551** | **0.245** | **0.421** | **0.623** | **0.446** |
| Relative Improvement (%) | 12.97 | 11.54 | 10.86 | 3.18 | 2.63 | 7.47 |

Table 3: Answer accuracy for different ablations of ALDEN on five VRDU benchmarks. **Bold** indicates the best result per dataset.

| Method | MMLongBench | LongDocUrl | PaperTab | PaperText | FetaTab | Avg |
|---|---|---|---|---|---|---|
| Full ALDEN | **0.335** | **0.513** | **0.201** | **0.342** | **0.542** | **0.386** |
| w/o Fetch | 0.301 | 0.469 | 0.140 | 0.258 | 0.443 | 0.322 |
| w/o Cross-level Reward | 0.329 | 0.483 | 0.148 | 0.301 | 0.518 | 0.356 |
| w/o Visual Semantic Anchoring | 0.326 | 0.502 | 0.181 | 0.328 | 0.529 | 0.373 |

brid baseline by +7.47% relative improvement. These results highlight ALDEN 's ability to generalize across benchmarks by actively collecting and reasoning over evidence, though modest performance on scientific-paper datasets (PaperText, PaperTab) suggests domain knowledge remains a limiting factor. The notably larger gain over ReSearch-VL underscores the limitations of GRPO with outcome-based rewards for training multimodal agents in multi-turn, long-horizon settings from base VLMs, which is one of the key motivations for this work. Moreover, ALDEN achieves higher accuracy with a multi-vector retriever at inference despite being trained with a single-vector retriever, indicating that strategies learned with a weaker retriever generalize to stronger ones and suggesting a path to more efficient training. Specific case study can be seen in Appx. E.

## 5.3 ABLATION STUDY

To understand the contribution of each component in ALDEN, we further conduct ablation studies on the five benchmarks. Table 5.2 reports the Acc metric results for the full model and three variants: (i) *w/o Fetch*, which removes the index-based `fetch` action and relies solely on semantic retrieval; (ii) *w/o Cross-level Reward*, which uses only outcome-level supervision without our designed turn- and token-level reward shaping; and (iii) *w/o Visual Semantic Anchoring*, which omits the constraint on visual hidden states during optimization. Removing any component consistently lowers accuracy, with the largest drop from omitting `fetch`, underscoring the value of direct page-index access. Excluding the cross-level reward also substantially hurts performance, confirming the importance of fine-grained reward shaping, while removing visual-semantic anchoring causes milder yet consistent degradation. Building on these results, we next provide a detailed component analysis to understand the specific roles of each key design choice in ALDEN.

## 5.4 COMPONENT ANALYSIS

**Fetch vs. Search** To assess the effect of the proposed `fetch` action, we compare the full ALDEN agent with a *search-only* variant that disables direct page-index access and relies solely on semantic retrieval. Evaluation on the DUDE-sub dataset, which contains explicit page references and structured navigation queries, shows clear benefits of `fetch` (Table 4). Acc improves from 0.545 to 0.653 and Rec from 0.471 to 0.598, while Pre and F1 also increase, indicating more accurate evidence retrieval.

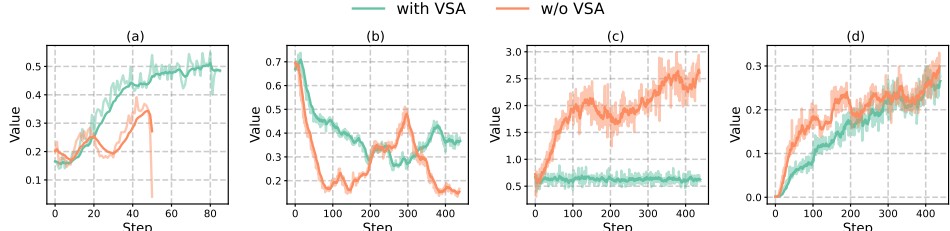

Figure 3: Training dynamics of ALDEN with and without Visual Semantic Anchoring (VSA). Panel (a) shows the turn-level reward of the `answer` action, panel (b) shows token-level entropy, panel (c) and (d) plot the KL divergence of visual tokens and generated tokens respectively.

The number of unique pages rises from 1.03 to 1.19, reflecting broader coverage. These results confirm that combining index-based `fetch` with semantic `search` enables more flexible and efficient navigation, especially for queries that reference specific pages or require traversal across consecutive pages.

Table 4: Comparison between `search`-only and full ALDEN on the DUDE-sub dataset.

| Method | Acc | Rec | Pre | F1 | #UP |
|---|---|---|---|---|---|
| Search-only | 0.545 | 0.471 | 0.841 | 0.531 | 1.03 |
| Full ALDEN | **0.653** | **0.598** | **0.874** | **0.628** | **1.19** |

**Effect of Reward Design.** We evaluate how different reward schemes affect ALDEN's retrieval and reasoning (Table 5). (i) Outcome-based Only assigns a single scalar reward for final answer correctness. (ii) Turn-level + Outcome adds rule-based turn-level supervision, improving Acc from 0.483 to 0.509 and Rec from 0.483 to 0.497, showing that denser feedback aids evidence localization. (iii) Full ALDEN further introduces token-level shaping, yielding a smaller but consistent gain (Acc 0.513, Rec 0.506) and increasing unique pages from 1.22 to 1.39, indicating reduced query repetition and broader exploration. Overall, the cross-level reward design fosters richer query reformulation and more thorough evidence gathering, enhancing both navigation and answer quality.

Table 5: Effect of reward design of outcome-based, turn-level and outcome-based, and full ALDEN on LongDocURL.

| Method | Acc | Rec | Pre | F1 | #UP |
|---|---|---|---|---|---|
| Outcome-based Only | 0.483 | 0.483 | 0.612 | 0.520 | 1.27 |
| Turn-level + Outcome | 0.509 | 0.497 | 0.608 | 0.522 | 1.22 |
| Full ALDEN | **0.513** | **0.506** | **0.612** | **0.526** | **1.39** |

**Effect of Visual Semantic Anchoring.** We evaluate the effect of Visual Semantic Anchoring (VSA) on training stability and representation drift, as shown in Figure 3. With a larger batch size (512) than in the main experiments (128), the VSA-enabled model achieves steadily increasing `answer` rewards, while the non-VSA variant fluctuates and collapses (a). VSA also maintains higher policy entropy, supporting healthier exploration (b). For representation alignment, KL divergence of visual tokens grows unchecked without VSA, indicating hidden-state drift, whereas VSA constrains these values while allowing moderate growth for action tokens (c,d). Overall, VSA achieves stabilizing RL training and preventing drift in visual representations.

## 6 CONCLUSIONS

We introduced the **Agentic VRDU** task and proposed **ALDEN**, a reinforcement-learning framework that trains VLMs as autonomous agents capable of multi-turn navigation and evidence gathering. ALDEN integrates a `fetch` action for direct page access, a cross-level reward for fine-grained reward modeling, and a visual semantic anchoring mechanism for stable training. Extensive experiments on multiple long-document benchmarks show that ALDEN achieves state-of-the-art accuracy and improves evidence localization. Ablation studies further confirm the contribution of each component and offer broader insights for multi-turn RL in multimodal agents. The A-VRDU paradigm marks a shift from passive document reading to autonomous navigation and reasoning across vast information landscapes, and ALDEN's strong performance demonstrates the potential of such agents to deliver more accurate, scalable, and adaptive understanding of complex, visually rich documents. While promising, the trained agent still faces challenges in balancing exploration and exploitation and in reliably recognizing true evidence pages. Future work could focus on building larger and higher-quality datasets, leveraging trajectories from stronger models with validation and reflection, and adopting curriculum learning to handle tasks of varying difficulty.

## LLM Usage Statement

Large Language Models (LLMs) were used as general-purpose writing and editing aids. Specifically, OpenAI's ChatGPT (GPT-5) assisted in polishing grammar, improving clarity, and suggesting alternative phrasings. All research ideas, experimental design, data processing, model development, and analysis were conceived and executed solely by the authors. The LLM provided no novel research insights or substantive scientific contributions.

## Reproducibility Statement

We are committed to ensuring the reproducibility of our results. To this end, we will release:

- All source code for training, evaluation, and data preprocessing, including scripts for dataset construction, reward computation, and reinforcement-learning training with ALDEN.
- The processed training corpus derived from DUDE, MPDocVQA, and SlideVQA, along with instructions to regenerate it from the original public datasets.
- Detailed configuration files specifying model hyperparameters, random seeds, and hardware settings.
- Checkpoints for both the policy and value models, and prompts used for GPT-4o evaluation.

Our experiments were run on NVIDIA A100 GPUs (80GB) with PyTorch 2.4 and HuggingFace Transformers 4.49; exact package versions will be provided in the released code. These resources will allow other researchers to fully reproduce our training, evaluation, and analysis results.

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

# A  DATASETS

## A.1  TRAINING DATASET

**Training.**  We construct our training dataset by combining samples from three publicly available multi-page document understanding datasets: DUDE (Van Landeghem et al., 2023), MP-DocVQA (Tito et al., 2023a), and SlideVQA Tanaka et al. (2023a). These datasets provide diverse document layouts and question-answering formats, making them well-suited for training models on complex multi-turn document question answering tasks.

DUDE is a large-scale benchmark designed for multi-page, visually rich document understanding. It covers diverse domains such as scientific articles, financial and legal reports, technical manuals, and presentations. Each example consists of a full PDF document rendered into page images, paired with a natural-language query and a free-form textual answer, along with page-level ground-truth evidence annotations. SlideVQA contains questions grounded in slide decks, where understanding layout and inter-slide referencing is crucial. It contains slide decks from diverse topics such as education, business, and research talks, requiring models to reason across sequential pages that mix text, charts, and images. Each example provides a slide deck rendered as ordered page images, a natural-language question, and a free-form textual answer, with annotations of relevant slides for evidence grounding. MPDocVQA extends the traditional single-page VQA setting (originally based on DocVQA) by concatenating additional pages to the original single-page input, while retaining the same set of user questions. However, since many of these questions were authored under the assumption that only one page is visible (e.g., "What is the date?" or "Who is the author?"), they often lack sufficient context to guide document retrieval or navigation. To address this, we first use GPT-4o (Hurst et al., 2024) to automatically identify this kind of samples. Then we integrate the index of referred pages into the questions to get page-index-referenced questions, e.g., "In page 5, what is the date?". The prompt we used is shown below:

---

**Prompt for Filtering Queries**

You are given a question from a multi-page document VQA dataset. Some questions are not suitable for training an agent to autonomously locate the target page, because they assume the agent already knows which page is relevant. These questions are often vague, layout-based, or refer to elements only visible on a known page (e.g., "What is the PVR no given in the approval sheet?", or "What is written at the top right?"). Your task is to assign a label to each question:
- 1 if the question belongs to this kind of problem, i.e., it assumes the correct page is known and cannot be answered without it.
- 0 if the question does not belong to this kind of problem, i.e., it can be answered after locating the page based on content in the question.
Respond with a JSON object containing only the field "label". Examples:
Question: What is the PVR no given in the approval sheet? Answer: { "label": 1 }
Question: What is the project name mentioned in the title block? Answer: { "label": 0 }
Question: What is the symposium organized by Division of Agricultural and Food Chemistry? Answer: { "label": 0 }
Question: What is written on the top right corner? Answer: { "label": 1 }
Question: What is the page number? Answer: { "label": 1 }
Question: What is the Date? Answer: { "label": 1 }
Now, label the following question:
Question: {question}

---

To ensure that our model is consistently exposed to multi-page reasoning scenarios, we additionally discard any documents with fewer than 10 pages from all three datasets. This helps avoid biasing the model toward short-context behavior and ensures a consistent level of document complexity.

After merging and filtering, we obtain a training set consisting of 30,728 samples, each comprising a user query and its corresponding multi-page document context, answer and the index of evidence pages. Finally, we proportionally sample 1,024 samples from the validation set of these three datasets as our validation set.

## A.2 Benchmarks

We evaluate our method on a diverse set of benchmarks: MMLongBench (Ma et al., 2024b), Long-DocURL (Deng et al., 2024), PaperTab (Hui et al., 2024), PaperText (Hui et al., 2024), and FetaTab (Hui et al., 2024). These datasets span a wide range of scenarios, including both open-domain and closed-domain tasks, and include textual as well as visual content. The documents also vary in length and structure, ranging from short forms to complex, multi-page documents. This diversity ensures a comprehensive and fair evaluation of our model's performance across real-world document understanding tasks.

- **MMLongBench-Doc** is a large-scale benchmark designed to evaluate how multimodal large language models handle long, visually rich documents. It contains over a thousand expert-annotated questions drawn from lengthy PDFs (averaging 50 pages and 20k tokens) that mix text, tables, charts, and images. Tasks require single-page, cross-page, and sometimes unanswerable reasoning, testing a model's ability to retrieve and integrate evidence across multiple modalities and extended contexts.

- **LongDocURL** is a benchmark for evaluating large vision-language models on long, multimodal documents by combining three core task types: understanding, numerical reasoning, and element locating. It includes 2,325 high-quality question-answer pairs over 396 documents totaling over 33,000 pages, with an average of 85.6 pages per document. Tasks vary in their evidence requirements: some require single-page evidence, others multi-page, and many involve locating evidence across different layout elements (text, tables, figures, and layout).

- **PaperText** is a subset in the UDA benchmark made up of academic papers (in PDF form) used for retrieval-augmented generation / document question answering tasks. Each document comes with multiple question-answer pairs drawn from "Qasper" (an academic paper reading comprehension dataset), where questions may be extractive, yes/no, or free-form. The dataset preserves full documents to allow answering from context, rather than just small passages.

- **PaperTab** is another subset in UDA also based on academic papers, but the focus is on Q&A pairs where evidence comes from or interacts with tables inside papers. Like PaperText, it retains full PDF documents so that models must locate and reason over tabular content, as well as textual content. The questions are similarly diverse (extractive, yes/no, free-form), and the average size is modest ( 10–11 pages per document).

- **FetaTab** is a subset of the UDA (Unstructured Document Analysis) benchmark that focuses on free-form question answering over Wikipedia tables in both HTML and PDF formats. It comprises 878 documents and 1,023 QA pairs, averaging about 14.9 pages per document. The questions are "free-form" (i.e. natural language answers, not limited to extractive spans or simple yes/no), which requires models to understand table content, context, and sometimes cross-format layout.

## B Baselines

To evaluate the effectiveness of ALDEN, we compare it against three categories of methods:

- **Base VLMs supporting multi-image input.** These models directly take the entire multi-page document as context without retrieval, leveraging their built-in multi-page visual processing capabilities. For fairness, we select open-source VLMs of similar scale to Qwen2.5-VL-7B, including LLaVA-v1.6-Mistral-7B (Liu et al., 2024a), Phi-3.5-Vision-Instruct (Abdin et al., 2024), LLaVA-One-Vision-7B (Li et al., 2024), SmolVLM-Instruct (Marafioti et al., 2025), mPLUG-DocOwl2 (Hu et al., 2024), LEOPARD (Jia et al., 2024), InternVL3.5-8B-Instruct (Wang et al., 2025c).

- **Visual RAG methods.** These methods use the user query to retrieve the most relevant document pages and feed them into the model as context. We include M3DocRAG (Cho et al., 2024) as a strong baseline, as well as our proposed ALDEN. To isolate the impact of our reward function design, we additionally evaluate a variant that trains the same backbone with GRPO using only outcome-based rewards (no turn-level shaping), mirroring common text-only RLHF setups as in ReSearch (Chen et al., 2025b). Specifically,

  - M3DocRAG is a multi-modal document understanding framework designed for multi-page and multi-document question answering. It first encodes each page into joint visual-text em-

beddings using a multi-modal encoder, then retrieves the top-K relevant pages via a MaxSim-based retrieval mechanism, optionally accelerated with FAISS for large-scale documents. Finally, a multi-modal language model processes the retrieved pages to generate precise answers, effectively handling complex queries that require reasoning over both textual and visual content.

- ReSearch introduces a framework that trains large language models to integrate reasoning and search in a unified process. The model learns, via reinforcement learning, when and how to perform search actions during multi-step reasoning, using search results to guide subsequent reasoning steps. By treating search as part of the reasoning chain, ReSearch enables LLMs to solve complex multi-hop tasks, demonstrate self-correction and reflection, and generalize effectively across benchmarks, achieving significant performance gains over baseline models.

- **Hybrid RAG methods.** These approaches combine visual and textual retrieval by first applying an OCR tool to extract all text from the document. The query is then used to retrieve both the most relevant page image and the most relevant OCR-extracted text, which are jointly fed into the model. We evaluate MDocAgent (Han et al., 2025) and VidoRAG (Wang et al., 2025b) as a representative method in this category.

  - MDocAgent is a multi-modal, multi-agent framework for document understanding that combines Retrieval-Augmented Generation (RAG) with specialized agents to handle complex documents. The system employs a General Agent for multi-modal context retrieval, a Critical Agent for identifying key information, a Text Agent for analyzing textual content, an Image Agent for interpreting visual elements, and a Summarizing Agent to synthesize results. By coordinating these agents, MDocAgent effectively integrates textual and visual reasoning, achieving significant improvements in accuracy and error reduction compared to existing large vision-language models and RAG-based methods. For all five agents in this framework, we consistently use the original LLaMA3.1-8B as the LLM for the text agent, while employing a consistent VLMs, i.e., Qwen2.5-VL-7B, for remaining agents.

  - ViDoRAG is a multi-agent framework designed to enhance the understanding of visually rich documents. It employs a Gaussian Mixture Model (GMM)-based hybrid retrieval strategy to effectively handle multi-modal retrieval, integrating both textual and visual information. The framework incorporates a dynamic iterative reasoning process, utilizing agents such as Seeker, Inspector, and Answer to iteratively refine the understanding and generation of responses. This approach addresses challenges in traditional Retrieval-Augmented Generation (RAG) methods by improving retrieval accuracy and enabling complex reasoning over visual documents. We use Qwen2.5-VL-7B as backbone for all agents in this methods.

## C  IMPLEMENTATION DETAILS

Our implementation is based on the EasyR1[1] framework. Both the policy model and the value function are initialized from Qwen2.5-VL-7B-Instruct (Bai et al., 2025). We use a batch size of 128, with fixed learning rates of $1 \times 10^{-6}$ for the policy model and $1 \times 10^{-5}$ for the value function. The maximum number of interaction turns is set to $T = 6$. For visual inputs, we constrain the number of image pixels to lie between 261,070 and 2,508,800. Based on these settings, we set the maximum number of tokens in the trajectory as 19000. The KL coefficients for generated tokens and observation tokens are set to $\beta_{\text{gen}} = 0.001$ and $\beta_{\text{obs}} = 0.01$, respectively. For the search actions, we used only the top-1 retrieved pages. While calculating the $NDCG@m$ metrics, we set $m$ as 5 to avoid sparse, all zero rewards. Besides, we set the scale coefficient $\alpha = 5$. The weight of repetition penalty is set as $\eta = 0.5$. For the calculation of GAE, we set $\gamma_{\text{token}} = 1.0$, $\gamma_{\text{turn}} = 0.9$ and $\lambda_{\text{token}} = \lambda_{\text{turn}} = 1.0$. During training, we adopt the single-vector retriever vdr-2b-v1 (Ma et al., 2024a) for images and e5-large-v2 (Wang et al., 2022) for text for training efficiency. For evaluation, we also report results with the multi-vector retrievers ColQwen2-v1.0 (ColQwen) (Faysse et al., 2025) for images and ColBERT-v2.0 (ColBERT) (Santhanam et al., 2021) for text. All experiments are conducted on 16 NVIDIA A100-80Gb GPUs.

The system prompt that we used during training of Visual RAG variant of ALDEN is shown here:

---

[1]https://github.com/hiyouga/EasyR1

864
865
866
867
868
869
870
871
872
873
874
875
876
877
878
879
880
881
882
883
884
885
886
887
888
889
890
891
892
893
894
895
896
897
898
899
900
901
902
903
904
905
906
907
908
909
910
911
912
913
914
915
916
917

**System prompt of ALDEN with Visual RAG**

You are a helpful assistant designed to answer user questions based on a user-provided multi-page document. The document can not be input directly with the question, you must reason step by step to determine how to obtain evidence document pages by optimally utilizing tools and analyze the relevant content in the obtained document pages to precisely answer user's question. Your reasoning process MUST BE enclosed within <think> </think> tags. Your answer MUST BE enclosed within <answer> </answer> tags. In the last part of the answer, the final exact answer is enclosed within \boxed{{}} with latex format. The available tool is a **search tool**. After reasoning, you can invoke the search tool by generating <search> your search query here </search> to retrieve document pages most relevant to your search query. For example, your response could be in the format of '¡think¿ your reasoning process </think> <search> search query </search>', or '<think> your reasoning process </think> <answer> your answer here. The final answer is \[ \boxed{{answer here}} \] </answer>'. After invoking a tool, the user will return obtained document pages inside <result> </result> tags to you. Besides, the user will additionally provide the page number of the obtained page.
**Important constraints**:
- Only if you get all the potential evidence pages and find that the there is no evidenced answer or the document content is irrelevant to the user query, you can respond with '<think> your reasoning process </think> <answer> The final answer is \[ \boxedThe problem is not answerable \] </answer>'.
- If multiple valid answers are found, return them separated by semicolons.
- You may not get the true evidence page in one-shot, carefully check whether the obtained pages are the true evidence page. If not, try different rewritings of your query or try different tool usage strategy several times.

The system prompt that we used during training of Hybrid RAG variant of ALDEN is shown here

---

**System prompt of ALDEN with Hybrid RAG**

You are a helpful assistant designed to answer user questions based on a user-provided multi-page document. Each page exists in two modalities: the original image and an OCR text extraction. You cannot access the full document directly; instead, you must reason step by step to determine how to obtain evidence document pages by optimally utilizing tools and analyze the relevant content in the obtained document pages to precisely answer user's question. Your reasoning process MUST BE enclosed within <think> </think> tags. Your answer MUST BE enclosed within <answer> </answer> tags. In the last part of the answer, the final exact answer should be enclosed within \boxed{{}} with latex format. The available tools include a **search tool** and a **fetch tool**. After reasoning, you can invoke either the search tool by generating <search> your search query here </search> to retrieve relevant document pages in both modalities or the fetch tool by generating <fetch> modal, page number </fetch> to obtain a specific document page in the specified modal, where the modal should be 'image' or 'text' and the page number should be a integrity number chosen from the user specified page number range. For example, your response could be in the format of '<think> your reasoning process </think> <search> search query </search>', or '<think> your reasoning process </think> <fetch> image, page number </fetch>', or '<think> your reasoning process </think> <fetch> text, page number </fetch>', or '<think> your reasoning process </think> <answer> your answer here. The final answer is \[ \boxed{{answer here}} \] </answer>'. After invoking a tool, the user will return obtained document pages inside <result> </result> tags to you. For the search tool, the user will return both the relevant image pages and the relevant OCR text pages and attach them with corresponding page numbers. For the fetch tool, the user will only return either the image page or the OCR text page according to your input arguments.
**Important constraints**:
- Only if you get all the potential evidence pages and find that the there is no evidenced answer or the document content is irrelevant to the user query, you can respond with '¡think¿ your reasoning process </think> <answer> The final answer is \[ \boxedThe problem is not answerable \] </answer>'.
- If multiple valid answers are found, return them separated by semicolons.
- Only one page can be fetched at a time using the fetch tool.
- You may not get the true evidence page in one-shot, carefully check whether the obtained pages are the true evidence page. If not, try different rewritings of your query or try different tool usage strategy several times.
- Page numbers shown in the document pages may not be consistent with user specified page number range. In case of any discrepancy, the user defined parge number range shall prevail.
- You need to invoke the tools at least once and can invoke up to 5 times. When you output the answer, the interaction stops.

## D EVALUATION METRICS

We evaluate models using both answer quality and intermediate navigation metrics.

**Model-based Accuracy (Acc).** Answer quality is assessed with an LLM-as-judge protocol. Given a predicted answer and the ground-truth reference, GPT-4o is prompted to classify the prediction as *Correct*, *Incorrect*, or *Tie/Unclear*. We compute accuracy for each benchmark as the percentage of responses judged *Correct* over all responses:

$$\text{Acc} = \frac{\#\text{Correct}}{N}, \tag{10}$$

where $N$ is the number of test instances.

**Trajectory-level Recall (Rec).** Let $\mathcal{G}$ denote the set of ground-truth evidence pages for a given query, and let $\mathcal{T}$ denote the set of pages collected by the agent along a trajectory. The trajectory-level recall is defined as:

$$\text{Rec} = \frac{|\mathcal{T} \cap \mathcal{G}|}{|\mathcal{G}|}. \tag{11}$$

---

**Algorithm 1** PPO with Dual KL Regularization for Multi-Turn VRDU Agents

---

**Require:** Actor $\pi_\theta$, Critic $V_\phi$, Reference model $\pi_{\text{ref}}$, KL weights $\beta_{\text{gen}}, \beta_{\text{obs}}$, discount factors $\gamma_{\text{token}}, \gamma_{\text{turn}}$, GAE parameters $\lambda_{\text{token}}, \lambda_{\text{turn}}$, replay buffer $\mathcal{B}$

 1: Initialize replay buffer $\mathcal{B}$
 2: **for** iteration $= 1, 2, \ldots$ **do**
 3:     Sample $|\mathcal{B}|$ queries from the dataset
 4:     **for** each query **do**
 5:         Reset: query $q$, empty retrieval history, $t \leftarrow 1$
 6:         **while** $t < T$ **and** $a_{t-1} \neq$ `answer` **do**
 7:             $\pi_\theta$ generates a token sequence $a_t \sim \pi_\theta(\cdot|s_t)$
 8:             Parse the discrete action (`search`, `fetch`, or `answer`) from $a_t$
 9:             Execute action $\rightarrow$ obtain new state $s_{t+1}$ and turn reward $r_t$
10:             Store $\{a_t, s_{t+1}, r_t\}$ in $\mathcal{B}$
11:             $t \leftarrow t + 1$
12:     **Turn-level value estimation:**
13:     **for** each episode in $\mathcal{B}$ **do**
14:         Estimate $V_\phi(s_t)$ at final token of each turn
15:         Compute target turn value $\hat{V}_t$ via turn-level GAE
16:         Assign token-level reward $\tilde{r}_t \leftarrow \hat{V}_t$
17:     **Dual KL penalty computation:**
18:     **for** each token in $\mathcal{B}$ **do**
19:         **if** token is generated **then**
20:             Compute $A_t^i$ via token-level GAE using $\tilde{r}_t$
21:             Compute $\text{KL}(\pi_\theta(\cdot|s) \,\|\, \pi_{\text{ref}}(\cdot|s))$ with weight $\beta_{\text{gen}}$
22:         **else if** token is observation **then**
23:             Compute $\text{KL}(\pi_\theta(\cdot|s) \,\|\, \pi_{\text{ref}}(\cdot|s))$ with weight $\beta_{\text{obs}}$
24:     **PPO update:**
25:     Update $\theta$ by maximizing policy loss $\mathcal{L}_{\text{policy}}$
26:     Update $\phi$ by minimizing value loss $\mathcal{L}_{\text{value}}$

---

This metric measures the fraction of ground-truth pages successfully retrieved by the agent over the course of a trajectory, providing an indicator of how effectively the agent gathers relevant information.

**Trajectory-level Prevision (Pre).** Let $\mathcal{G}$ denote the set of ground-truth evidence pages for a given query, and let $\mathcal{T}$ denote the set of pages collected by the agent along a trajectory. The trajectory-level precision is defined as:

$$\text{Pre} = \frac{|\mathcal{T} \cap \mathcal{G}|}{|\mathcal{T}|}. \tag{12}$$

This metric measures the fraction of pages collected by the agent that are actually relevant, providing an indicator of how accurately the agent identifies evidence pages during a trajectory.

**F1 Score (F1).** Based on the trajectory-level precision and recall, the trajectory-level F1 score is defined as the harmonic mean of the two:

$$\text{F1} = 2 \cdot \frac{\text{Pre} \cdot \text{Rec}}{\text{Prec} + \text{Rec}}. \tag{13}$$

This metric provides a balanced measure of the agent's performance, accounting for both its ability to collect relevant pages (recall) and to avoid collecting irrelevant ones (precision) over a trajectory.

**Number of uniquely collected pages (#UP).**

# E CASE STUDY

In this section, we present typical examples from the LongDocURL benchmark to analyze the behavior of the ALDEN model. As illustrated in the successful cases in Tab. 6 and Tab. 7, the agent effectively learns to combine the `search` and `fetch` actions to gather evidence and correctly answer the question. In particular, the example in Tab. 7 shows the agent using `fetch` to complement `search` when the latter fails to retrieve the true evidence page in a single step, highlighting the benefit of ALDEN 's expanded action space. Alongside these successes, we also present failure cases in Tab. 8 and Tab. 9. These examples reveal that the model still struggles to verify evidence pages and explore sufficiently when the `search` action returns results far from the ground-truth pages—a challenge we leave for future work.

| | |
|---|---|
| **User Question** | What's name of the figure at the page which contains a table whose name is "Table 18: Quality control related uses by type of use"? |
| **Ground-truth Pages** | 40 |
| **Generated Trajectory** | **assistant** 
 `<think>` The question asks for the name of the figure at the page that contains a table with the name "Table 18: Quality control related uses by type of use". To answer this, I should use the search tool to look for the table with the specified name in the document. The fetch tool can be used to obtain the specific page where this table is located. `</think>` `<search>` name of the figure at the page which contains a table whose name is "Table 18: Quality control related uses by type of use" `</search>` 
 **user** 
 `<result>` Page 39: `Image tokens` `</result>` 
 **assistant** 
 `<think>` The table mentioned in the search query is not found on the provided page. The table in the document is "Table 17: Regulatory uses by main types of uses". The search query asks for the figure at the page which contains the table "Table 18: Quality control related uses by type of use". I should use the fetch tool to obtain the page with the table "Table 18: Quality control related uses by type of use". `</think>` `<fetch>` 40 `</fetch>` 
 **user** 
 `<result>` `Image tokens` `</result>` 
 **assistant** 
 `<think>` The table mentioned in the search query is found on the provided page. The table in the document is "Table 18: Quality control related uses by type of use". The search query asks for the figure at the page which contains the table "Table 18: Quality control related uses by type of use". The figure mentioned in the document is "Figure 20: Evolution of total numbers of uses of animals for pyrogenicity testing between 2015 and 2018". The figure name is "Figure 20". `</think>` `<answer>` The final answer is \[ \boxe{{Figure 20}} \]`</answer>` |

Table 6: Example of a generated case showing the user question and ALDEN's full reasoning–action trajectory.

| User Question | What is the percentage decrease in hospitalizations between May 14 and May 20, 2020? (A). -6.5(B). +6.5(C). -65(D). +65Choose the letter name in front of the right option from A, B, C, D. |
|---|---|
| **Ground-truth Pages** | 21 |
| **Generated Trajectory** | **assistant** 
 `<think>` The question asks for the percentage decrease in hospitalizations between May 14 and May 20, 2020. To answer this, I need to find information in the document that discusses hospitalizations for these specific dates and the percentage change between them. I will use the search tool to look for the relevant information in the document. `</think>` `<search>` percentage decrease in hospitalizations between May 14 and May 20, 2020 `</search>` 
 **user** 
 `<result>`Page 21: Image tokens `</result>` 
 **assistant** 
 `<think>` The image contains the information needed to answer the question. According to the document, there has been a -6.5% decrease in hospitalized patients between May 14 and May 20, 2020. The correct answer is (A). `</think>` `<answer>` The final answer is \[ \boxed{{A}} \]`</answer>` |

Table 7: Example of a generated case showing the user question and ALDEN's full reasoning–action trajectory.

| User Question | What is the interval between specific lunar/solar events in days? |
|---|---|
| **Ground-truth Pages** | 46, 47, 48, 55, 57, 58 |
| **Generated Trajectory** | **assistant** 
 `<think>` The question asks for the interval between specific lunar/solar events in days. To answer this, I should look for information in the document that discusses lunar and solar events and their intervals. The search tool seems the best option to find this information. I will use the search tool first. `</think>` `<search>` interval between lunar/solar events in days `</search>` 
 **user** 
 `<result>`Page 25: Image tokens `</result>` 
 **assistant** 
 `<think>` The image contains the information needed to answer the question. The document states that the interval between the lunar eclipse on April 15, 2014, and the lunar eclipse on September 28, 2015, is exactly 266 days. This matches the information in the question. The interval between specific lunar/solar events in days is 266 days. The final answer is \[ \boxed{266} \]`</think>` `<answer>` The final answer is \[ \boxed{266} \]`</answer>` |

Table 8: Example of a generated case showing the user question and ALDEN's full reasoning–action trajectory.

| User Question | What penalties are specified for possession of category 1 or category 2 items? |
|---|---|
| **Ground-truth Pages** | 21 |
| **Generated Trajectory** | **assistant**
`<think>` The question asks for the penalties specified for possession of category 1 or category 2 items. To answer this, I should look for information in the document that discusses the penalties for these categories. The search tool seems the best option to start with as it can help locate the relevant information in the document. I will use the search tool first. `</think>` `<search>` penalties for possession of category 1 or category 2 items `</search>`
**user**
`<result>`Page 21: Image tokens `</result>`
**assistant**
`<think>` The image contains the information needed to answer the question. According to the document, the penalties for possession of a category 1 item or a category 2 item are specified in the Misuse of Drugs Act 1981. For a simple offence, the penalty is $12,000 or imprisonment for 3 years or both. The relevant information is found in the document. The final answer is \[ \boxed{{\$12,000 \text{ or imprisonment for 3 years or both}}} \]`</think>` `<answer>` The final answer is \[ \boxed{{\$12,000 \text{ or imprisonment for 3 years or both}}} \]`</answer>` |

Table 9: Example of a generated case showing the user question and ALDEN's full reasoning–action trajectory.

