# OpenReview forum: "ALDEN: Reinforcement Learning for Active Navigation and Evidence Gathering in Long Documents"
_ICLR.cc/2026/Conference — ICLR 2026 Conference Withdrawn Submission_

### Official Review · Reviewer_cadj · 2025-10-19

**Soundness:** 3
**Presentation:** 2
**Contribution:** 3
**Rating:** 4
**Confidence:** 3

**Summary:**

This paper tackles the significant and challenging problem of enabling Vision-Language Models (VLMs) to effectively process long, multi-page, visually-rich documents (VRDs). The authors convincingly argue that existing passive approaches, such as fixed Retrieval-Augmented Generation (RAG) pipelines, are limited in efficiency and their ability to generalize.
To address this, the paper introduces the "Agentic VRDU" (A-VRDU) task paradigm and presents ALDEN (Active Long-Document
Navigation), a reinforcement learning (RL) framework that fine-tunes VLMs to act as interactive agents capable of active navigation.

**Strengths:**

- Clear and Significant Research Problem: The shift from passive document understanding to an active "Agentic VRDU" paradigm is a timely, well-motivated, and valuable research direction. The paper clearly articulates the limitations of existing methods, such as the rigidity of fixed RAG workflows , providing a strong justification for an active, agent-based approach.
- Excellent Problem-Solution Mapping: The authors clearly identify three specific challenges in applying RL to A-VRDU: (1) the insufficiency of semantic-only retrieval for structured references , (2) the difficulty of multi-turn interaction with sparse rewards , and (3) training instability from the high dimensionality of visual tokens. They map these challenges directly to their three core solutions (fetch action, cross-level reward, and VSA), creating a very clear and compelling narrative.

**Weaknesses:**

- The fetch action seems to only be able to index a single-page document. However, in actual scenarios, the solution to the problem relies on multi-page documents (even discontinuous pages), and ALDEN cannot handle this situation.
- The cross-referencing of the tables seems a little inconsistent.
- The cross-level reward function, while effective, is complex and introduces numerous components (format reward, F1, NDCG, proximity distance, repetition penalty) and their associated hyperparameters (e.g., $\alpha$, $\eta$, $m$, $n$-grams). This could make the reward function brittle, difficult to tune, and potentially reduce its generalizability to different tasks or domains. A sensitivity analysis of these hyperparameters is notably absent.

**Questions:**

See Weaknesses.

---

### Official Review · Reviewer_RshX · 2025-10-28

**Soundness:** 2
**Presentation:** 3
**Contribution:** 2
**Rating:** 4
**Confidence:** 4

**Summary:**

The paper describes an approach for document visual question answering based on a multi-step reasoning process guided by a reinforcement learning strategy. The multi-step reasoning process is an extension of RAG-based approaches, and consists of one or more steps to retrieve relevant information across all the pages of the document and a final step of answering the question based on the retrieved evidence. In addition to traditional search based on semantic similarity among the query and the document content, the proposed approach integrates a index-based search based on the page number. With respect to the application of reinforcement learning to guide the reasoning process, the proposed method integrates both turn-level and token-level rewards to discourage repeated query formulations. The proposed method is evaluated on standard benchmarks for document understanding showing competitive results and showing the contribution of the different components of the framework.

**Strengths:**

- The paper shows how reinforcement learning can be used in an effective way in the design of a framework for document understanding using a multi-step reasoning process. The proposed framework achieves state-of-the-art results in standard benchmarks.
- The ablation study shows a positive impact in the performance of the model of the three new contributions of the paper: the fetch action, the cross-level reward function, and the visual semantic anchoring.

**Weaknesses:**

- Although the fetch action is shown to be useful to improve the performance, it seems to me that this is is mainly due to the bias of existing datasets towards questions asking about specific pages (an effect that is even stronger in the results of table 4 with the specific DUDE-sub dataset). In the more general and real DocVQA case, questions are expected to be mainly posed over the content of the document more than specific locations of the document, and thus, page numbers would probably have less relevance, and a fetch action on page number would have less relevance. In any case, in a more general case, we could expect a fetch action to recover not only pages, but other elements of the document such as captions, sections and subsections, tables or figures.
- I miss a more fine-grained analysis of the results according to different types of questions or evidence sources. At least some of the datasets (for instance, MMLongBench) provide annotations about the type of question (single-page, cross-page or unanswearable) and the type of evidence required to answer the question (paragraph, table, figure, ...). Showing a more fine-grained detail of results according to these categories would help to understand, for instance, if the overall improvement comes only from a better retrieval in single-page questions or also from a better multi-step reasoning in cross-page questions.
- From the results, it is not clear to me if the RL based multi-step reasoning process performs better than simple RAG-based methods. Table 3 shows that removing the fetch action, i.e, using only the search action, the average accuracy is 0.322. This setting is pretty much similar to RAG-based methods where only the search action is used. Table 2 shows that M3DocRAG achieves 0.378 and MDocAgent, 0.415, higher than ALDEN using only search, which could lead to the conclusion than simple RAG performs better than the RL framework when using only the search action. However, these results are not directly comparable since in table 3 ALDEN is combined with vdr-2b-v1 for retrieval while in table 2 M3DocRAG is used with ColQwen and MDocAgent with ColBERT+ColQwen. A direct comparison using the same retrieval methods would help to clarify this issue.

**Questions:**

- How many average search/fetch/answer steps are necessary to get the answer? Knowing this number would help to better understand the difference between the proposed framework and standard RAG-based methods, and see if several reasoning steps are really executed.

---

### Official Review · Reviewer_WR4t · 2025-10-29

**Soundness:** 2
**Presentation:** 3
**Contribution:** 3
**Rating:** 4
**Confidence:** 4

**Summary:**

This paper introduces ALDEN, a reinforcement learning framework that enables visual-language models to actively navigate and retrieve evidence from documents. Through novel action design (search, fetch, answer), cross-level rewards, and visual-semantic anchoring, ALDEN achieves strong state-of-the-art results on multiple long-document understanding benchmarks.

**Strengths:**

The paper offers a fresh perspective by framing long-document understanding as an active navigation problem. The introduction of explicit fetch actions for page-indexed retrieval is clever and bridges structured document navigation with multimodal reasoning. The technical implementation is well-motivated, especially the cross-level reward and the VSA mechanism, which stabilizes visual representations during RL training and effectively prevents multimodal collapse.

**Weaknesses:**

1. Both M3DocRAG and MDocAgent originally use ColPali as their default image retriever, but Table 2 reports results with ColQwen. Moreover, ALDEN uses Qwen2.5-VL, while M3DocRAG uses Qwen2-VL, and MDocAgent combines Llama3.1 and Qwen2-VL. These backbone differences make the comparison less fair. Including ablations where all methods use the same backbone would make the improvements more convincing.
2. ALDEN is a trained RL system that also performs multi-step reasoning at test time, while baselines are all untrained models and frameworks. This makes it unclear whether the gain comes from the RL training or the multi-step inference structure.
It would help to evaluate (a) untrained models under ALDEN’s multi-turn reasoning protocol, (b) other training methods (SFT, GRPO... instead of ALDEN's ablations) under ALDEN’s multi-step setup, to isolate the true source of improvement.

**Questions:**

1. Were M3DocRAG and MDocAgent evaluated with ColQwen retrievers at test time? Please clarify the retriever settings.
2. Could you add ablations using identical backbones across systems to ensure a fair comparison?
3. Could you test an untrained Qwen 2.5 VL, as well as trained ones using other training methodologies, under the same multi-step reasoning protocol?

Suggestion: Adding a case study would be helpful to better understand ALDEN’s inference process.

---

### Official Review · Reviewer_6qUY · 2025-10-31

**Soundness:** 4
**Presentation:** 3
**Contribution:** 3
**Rating:** 6
**Confidence:** 5

**Summary:**

This paper propose ALDEN, an agent-style reinforcement learning framework for long and visually complex document understanding (A-VRDU task). The paper expands the action space by introducing a page-index-based fetch action in addition to traditional semantic retrieval, designs cross-level rewards (combining turn-level and token-level signals) to alleviate the sparse reward issue, and proposes a visual-semantic anchoring mechanism(dual-path KL regularization) to stabilize RL training with a large number of visual tokens. The model is trained on a synthetic dataset derived from DUDE, MPDocVQA, and SlideVQA, and demonstrates significant performance improvements on five long-document benchmarks, with comprehensive ablation studies and training visualizations.

Overall, this is a potentially impactful work with a well-structured method that advances vision-language models from passive reading to active retrieval and reasoning. However, several theoretical and experimental clarifications are still needed in the rebuttal.

**Strengths:**

- Introducing an RL agent-style framework for long-document visual understanding (A-VRDU) is novel and of clear research significance.
- The integration of action space, reward modeling, and stabilization mechanisms forms a coherent and well-engineered system.
- The proposed page-index-based action is particularly effective for structured documents requiring explicit page referencing and sequential reasoning.
- The combination of turn-level and token-level rewards helps alleviate the sparsity and repetition issues in RL-based language generation.
- Applying dual KL constraints to both visual observation and text generation paths provides a promising solution to instability caused by dense visual tokens.
- The model achieves substantial improvements across multiple benchmarks, supported by ablation analyses that enhance result credibility.

**Weaknesses:**

- **Dataset issue**: Dataset is not public available, which limits its contribution.
- **Theoretical Novelty and Assumptions** – Does the paper provide any theoretical analysis of the visual-semantic anchoring mechanism’s convergence or optimization properties? For instance, does the dual-KL regularization affect the convergence guarantees of PPO? The paper currently presents only empirical evidence; a qualitative or theoretical discussion would strengthen the contribution.

- **Baseline Coverage and Generality** – The comparisons are primarily within the VRDU domain. The paper would benefit from comparisons with broader agentic or RL-for-VLM approaches

**Questions:**

- **Hyperparameter Sensitivity** – How are key hyperparameters (e.g., α for answer weighting, η for repetition penalty, β_gen, β_obs) selected? Are the results sensitive to these values? Please include hyperparameter tuning or sensitivity studies in the supplementary material.

---

### Note · Authors · 2025-11-25

I have read and agree with the venue's withdrawal policy on behalf of myself and my co-authors.